# Increased Elasticity Modulus of Polymeric Materials Is a Source of Surface Alterations in the Human Body

**DOI:** 10.3390/jfb12020024

**Published:** 2021-04-16

**Authors:** Matthias Kapischke, Igor Erlichman, Alexandra Pries

**Affiliations:** 1Department of Surgery, University Hospital Schleswig-Holstein, Campus Luebeck, Ratzeburger Allee 160, 23538 Luebeck, Germany; 2Department of Surgery, Klinikum Guetersloh, Reckenberger Strasse 19, 33332 Guetersloh, Germany; i.erlichman@gmx.de (I.E.); a.j.c.pries@gmail.com (A.P.)

**Keywords:** alloplastic material, hernia repair, structural alterations, NMR, FT-IR

## Abstract

The introduction of alloplastic materials (meshes) in hernia surgery has improved patient outcome by a radical reduction of hernia recurrence rate, but discussion about the biocompatibility of these implanted materials continues since observations of surface alterations of polypropylene and other alloplastic materials were published. This study intends to investigate if additives supplemented to alloplastic mesh materials merge into the solution and become analyzable. Four polypropylene and one polyester alloplastic material were incubated in different media for three weeks: distilled water, saline solution, urea solution, formalin, and hydrogen peroxide. No swelling or other changes were observed. Infrared spectroscopy scanning of incubated alloplastic materials and NMR studies of extracted solutions were performed to investigate loss of plasticizers. The surface of the mesh materials did not show any alterations independent of the incubation medium. FT-IR spectra before and after incubation did not show any differences. NMR spectra showed leaching of different plasticizers (PEG, sterically hindered phenols, thioester), of which there was more for polypropylene less for polyester. This could be the reason for the loss of elasticity of the alloplastic materials with consecutive physically induced surface alterations. A mixture of chemical reactions (oxidative stress with additive leaching from polymer fiber) in connection with physical alterations (increased elasticity modulus by loss of plasticizers) seem to be a source of these PP and PE alterations.

## 1. Introduction

The implantation of alloplastic materials such as polypropylene (PP) or polyester (PE) in hernia repair is currently the surgical standard [1]. It is incontestable that the use of these alloplastic materials in hernia repair is superior to non-mesh repair regarding the hernia recurrence and therefore the reoperation rate [2]. There is also the fact that these alloplastic materials, which are supposed to be clinically inert [3,4], are often described as showing structural changes [5,6,7,8]. The reasons of the surface alterations are assessed differently; they range from obviously water triggered swelling and modifications due to shrinking of the isolated PP fiber [5] to the impact of lysosomal enzymes to the formation of free radicals [6,8] and oxidative stress [9]. Conditions not common for the human body are often used as an explanation for the mesh alterations [10,11]. Moreover, the possibility that structural alterations are not the outcome of a chemical reaction but of a physical process has to be considered [3]. The aging processes of polymers is segmented into chemical and physical processes. It is difficult to differentiate both phenomenon, since these happen in parallel, and the resulting effects are overlapping [12]. Especially under the conditions in the human organism, the complexity of the process is not easily comprehended; traditionally, the chemical alteration process is the focus of interest. Within this complex occurrence, polymers undergo random chain scission, which induces embrittlement. Even though researchers had been mostly focused on chemical aspects of aging, it is noteworthy that a ductile–brittle transition can also occur without chemical degradation in polymers submitted to mechanical loads [13].

The here presented study is supposed to show that the released plasticizers from the alloplastic materials in cooperation with free radical reactions are a further reason for the structural alterations.

## 2. Materials and Methods

### 2.1. Meshes

Four different PP meshes and one PE mesh were used for investigations. Details are shown in Table 1. The focus here is PP meshes, since these meshes are commonly used in the national treatment reality. PE meshes are implanted much less often and therefore are employed as a comparator value in this study.

### 2.2. Investigation of Swelling and Surface Alteration

Pieces of the meshes described above—with the size of 1.0 cm × 1.0 cm—were incubated in different media to investigate swelling properties and surface alteration. As media, we used aqua bidest (self-produced in our laboratory), saline solution 0.9% (Braun), urea solution 50%, formalin 37%, and hydrogen peroxide (Merck Biosciences, Darmstadt Germany). Each deeper reaction within the alloplastic material is based on the uptake of aqueous solutions in most cases. As a logical consequence, this moisture absorption should accompany the “swelling” of PP fiber.

The incubation period was three weeks. Incubation conditions were pH 7.4 and 37.5 °C to simulate physiological conditions, with digital pictures taken before and after the incubation period. Fiber surface was evaluated with light microscopy. Since the absolute fiber thickness does not have an influence on the final statement, we determined the relative thickness (without decimal dimension) of 10 different fiber areas, which was outlined manually using Scion Image© vs. 4.0.3.2 (Scion Corp., Frederick, MD, USA). This measurement was done under consideration that absorption processes lead to swelling of the fiber. In case of multi-filament meshes, the thickness of the filament but not of the singular fiber was measured.

### 2.3. Infrared Spectroscopy (IR)

Fourier-transform infrared scans were performed to capture alterations of the material applying the attenuated total reflectance (ATR) method (Two FT-IR Spectrometer, PT-IR Polymer Resource Pack, Perkin Elmer, Rodgau, Germany).

### 2.4. Isolation of Extracts from Alloplastic Materials

Deuterated water and CDCl_3_ (deuterochloroform, Sigma-Aldrich, Munich, Germany) were used as polar and non-polar eluent, respectively.

For these investigations, pieces (5–6 g) of different materials (Hertra 2©, Parietene 3 PP1510©, Prolene PMN 3© and Surgipro Mesh© and Parietex Pes TEC1510©) were immersed in deuterated water for three weeks in dark laboratory glasses with grinded plugs. After removal of insoluble mesh, the solution was freeze-dried.

Pieces of the same batch of alloplastic hernia meshes were immersed in 10 mL chloroform in dark laboratory glasses with grinded plugs under nitrogen atmosphere. All glasses were stored in a dark surrounding for three weeks; incubation was performed at 37.5 °C. After this time, the solutions were investigated.

About 10–40 mg material were recovered and used with deuterated chloroform for further thorough characterization with Nuclear Magnetic Resonance (NMR) (^1^H, ^13^C, DEPT, H-H-COSY, HSQC and HMBC) as well as gas chromatography/mass spectrometry (GC/MS).

### 2.5. Characterization of the Extractable Material

^1^H and ^13^C NMR spectra were recorded on a Bruker Advance DRX 500 NMR spectrometer (Bruker Corp. Billerica, MA, USA). The ^1^H spectra recorded from D_2_O solutions of the water extractable materials were referenced on external 4.4-dimethyl-4-silapentane-1-sulfonic acid. ^1^H and ^13^C NMR spectra obtained from CDCl_3_ solution were referenced on internal tetramethylsilane (TMS) (Sigma-Aldrich, Munich, Germany). Both the whole extracts and the fraction after thin layer chromatography (TLC) separation were evaluated. H-H-COSY, HSQC, and HMBC correlation spectra were recorded to verify the signal assignment.

All experiments were blinded via a code number for the samples; unblinding was performed after final analysis.

### 2.6. Statistical Analysis

Statistical analysis was performed with Sigma Plot© Version 14.0 (Systat Inc., Erkrath, Germany); *t*-test was applied and *p*-values < 0.05 proved to be significant.

## 3. Results

### 3.1. Fiber Thickness

The measured thickness of the fibers after the three-week incubation period did not significantly deviate compared to the material before incubation (Table 2). We consider the very slight reduction of diameters in Hertra 2© within the measurement variance. The surface of the mesh materials did not show any alterations independent of the incubation medium.

### 3.2. Infrared Spectroscopy

None of the investigated meshes in the FT-IR spectroscopy showed any differences in the investigated alloplastic materials comparing mesh surfaces before and after the incubation period (representative results for three materials shown in Figure 1).

### 3.3. NMR Spectroscopy

NMR spectroscopy in water solution shows an exceedingly small amount (<100 µg) of extractable substances only; however, residues of oligomers of the PP were detectable.

The ^1^H NMR spectra of all CDCl_3_ extracts show groups of signals in the region of 1.7–0.8 ppm typical for PP-oligomers. As a representative example, the ^1^H spectrum of sample Hertra 2© is depicted in Figure 2. Corresponding signals are found in the ^13^C spectra at 38–18 ppm, and the mass spectra show peak cascades with distances of m/2 = 14 Da. In addition to the TMS peak (0 ppm) and the solvent peak (7.26 ppm), the ^1^H spectrum exhibits a group of signals around 3.7 ppm (with corresponding ^13^C signals around 70 ppm), which can be attributed to polyethylene glycol (PEG). The observation of the side signals from end group effects indicates that the molar mass of the PEG is rather low. In the HMBC spectrum (Figure 3), the signal of the outermost methylene group of the PEG (4.2 ppm) shows a correlation with a carbonyl-carbon (δ(^13^C) = 173 ppm), which in turn correlates with a triplet signal of an α-methylene signal of a carbon acid. Since the PEG signals show no correlation with a free OH-group, it can be concluded that both end groups of the PEG are esterified with carbon acids (Figure 2). Since the other signals of the carbon acids are superposed by the PP-signals, the type of fatty acid could not be identified; very often, oleic, stearic, or lauric acids are used. From the integral intensities of all PEG signals in relation to the α-methylene signal, a molar mass of the PEG of 400 Da is calculated. Polyethylene glycol fatty esters are used as emulsifiers, lubricants, dispersing agents, plasticizers in polymer films and coatings, and as antistatic agents. In polymer fiber production, they act as lubricant and antistatic agents to reduce friction and electrostatic charging during high-speed spinning [14,15]. They were found in all samples of meshes.

The ^1^H NMR spectrum of Hertra 2© (Figure 2) also shows several small peaks between 7.6 and 6.9 ppm, which originate from aromatic compounds. The HMBC spectrum (Figure 3) displays correlations of these aromatic protons with a few strong singuletts in the range of 1.45–1.25 ppm, which arise from tertiary-butyl substituents. Corresponding resonances occur in the ^13^C and HSQC spectra at around 30 ppm. In the ^1^H spectrum (Figure 2), a weak signal of a phenolic OH-group is detectable at 5.06 ppm, which exhibits only one J^3^-coupling with aromatic carbons (135 ppm) in the HMBC, thus indicating a symmetrical substitution pattern. Additional correlation signals reveal J^3^-couplings of the quaternary tertiary-butyl carbons with H-atoms in 3 and 5 position. Since no further correlations are observed, the C-4 position must be substituted by a heteroatom (e.g., sulfur).

This component is probably 4,4′-thiobis(di-2,6-di-tert-butylphenol) (Chemical Book No: 0356756). The sample contains other compounds of similar structure. In the ^1^H NMR spectrum of the sample Surgipro Mesh© (not shown), a singulett signal occurs at 2.3 ppm, which shows correlations with aromatic protons and tertiary-butyl signals in the HMBC. The substitution pattern and all correlations indicate the presence of a bis-tert-butyl cresol derivate, which is probably 4,4′-thiobis(6-tert-butyl-m-cresol) (CAS 96-69-5). These compounds belong to the class of sterically hindered phenols, which serve as antioxidizing agents in polyolefins [16].

They are utilized to stabilize polyolefins such as PEs, PP, or PS in cosmetic, medical, and food packing applications, since their toxicity is lower than comparable sterically hindered amine compounds [17]. The use of polyolefins in medical application is widely known [18].

The migrations of these types of antioxidants from PP into various liquids have been investigated and proved earlier [19,20].

The CDCl_3_ extract of sample Parietene 3 PP 1510© in general shows the same NMR spectra and contains consequently the same compounds, washed out of the PP meshes previously, indicating that the combination of retrieved additives (plasticizer, emulsifier, antioxidizer) are components of the raw polypropylene and not added during the mesh fiber production (not shown).

The ^1^H NMR spectra of all PP samples show exceedingly small resonance peaks at 0.1 ppm, which originate from silicon oil, which was probably used as a processing aid during the mesh fiber production.

In the CDCl_3_ extracts of samples Prolene PMN 3© and Surgipro Mesh©, more compounds could be identified. In the ^1^H spectrum of Prolene PMN 3© (Figure 4), three signals are clearly visible at 4.1, 2.8, and 2.6 ppm, which show correlations in the HMBC spectrum with resonances at 1.6, 1.3, and 0.9 ppm, resp., the latter being typical for fatty acid esters. By careful peak assignment, comparison with ^1^H and ^13^C reference spectra and mass spectroscopy, this compound could clearly be identified as dilauryl thiodipropionate (synonym: didodecyl-3,3′-thiodipropionat) (CAS 123-28-4) (Figure 4A), which is a secondary or auxiliary antioxidant exhibiting synergistic effects with the primary phenolic.

The NMR spectrum of the CDCl_3_ extract of sample Parietex PES TEC 1510© shows the typical resonance signals of ethylene terephthalate (δ(^1^H) = 8.1, 4.7 ppm, (δ(^13^C = 165.3, 133.8, 129.7, 62.8 ppm, Figure 5). The molecular ion peaks at M/z = 576 in the mass spectrum (Figure 5B) reveals that they occur mainly from the cyclic trimer tris(ethylene terephthalate) (Figure 5A). It is well known that cyclic oligomers are extractable from commercial polyethylene terephthalate (PET) to ≈1% and crystallize on PET surfaces upon heating [21,22].

## 4. Discussion

### 4.1. Alloplastic Materials and Foreign Body Reaction in the Host

The introduction of alloplastic materials such as PP and PES has accomplished Billroth’s vision: “If we could artificially produce tissue of the density and toughness of fascia and tendon, the secret of the radical cure of hernia would be discovered” [23]. The positive effect of these materials is an improvement of patient outcome in hernia repair by a radical reduction of recurrence rate [2]. In general, PP is considered as chemically inert [4]. Implantation of the alloplastic material leads to a host reaction, the so-called Foreign Body Reaction (FBR). FBR is the superordinate concept for a biomaterial induced cascade release of bio-reactive agents and invasion of immunocompetent cells [24]. This process intensity depends, beside other factors, on the kind of surface of the implanted material [25]. Buddy Ratner implemented the term “bio-tolerability” to describe the ability of a material to reside in the body for long periods of time with only low degrees of inflammatory reaction [26,27]. A long retention time in the human body is the goal of every hernia mesh.

The cascade for FBR described above, containing preferred macrophages and foreign body giant cells, produces a privileged environmental space around the surface of the alloplastic material [24]. The release of reactive oxygen intermediates, degradative enzymes, etc. in high concentration was called to account for this biodegradation process [28,29]. For several polyester biomaterials, these degradation processes were shown [24,30]. Interestingly, for the polyester subgroup polyethylene terephthalate, these degradation processes were described only rarely [24].

### 4.2. Swelling and FT-IR

It is well known that PP fibers, although being chemically mostly inert, take up aqueous liquids. This process is temperature-dependent and more distinct in temperatures above those of the human body. However, this has been described from temperatures of 23 °C upwards [31,32]. This moisture content has a relevant influence on the mechanical heaviness of PP. Our experimental approach (infrared scan and swelling measurements) did show here that aqueous solutions did not alter PP materials in our experiments, which includes also aggressive liquids such as urea and hydrogen peroxide. These findings are in accordance with former studies [3,9]. Swelling mechanisms primarily triggered by water intake appear to be limited relevant as well (Table 2). The intention of this approach was to exclude a relevant influence of swelling on the surface structure.

### 4.3. Oxidative Stress and Polymer Additives

In vitro models for investigation of the oxidative stress are quite old and have system deficits that are inherent to the model [33,34]. In this study, we wanted to emulate this aggressive-reactive environment by our selected incubation solutions (hydrogen peroxide, urea, deuterochloroform). NMR spectroscopy did show that emulsifiers, antioxidants, and plasticizers (PEG, thioester, hindered phenols) were able to wash out from the alloplastic material. Primary antioxidants are so-called H-donators or radical scavengers; as light abstracted H-atoms, they are supposed to protect the polymer chain. The proven sterically hindered phenols are good H-donators. Secondary antioxidants (for example thioester) may not play a role in the human body in their original significance as a high-temperature process stabilizer; given that these are present, they may have the known synergistic effects [35].

The good antioxidants should be distributed in the PP in a homogenous fashion and show low volatility. However, consumption on the surface leads to an unequal distribution and loss from the PP fiber, a missing stabilization of the PP, and finally to (physical and/or chemical) aging of the polymer. The same has been proven for other additives [36]. Our hypothesis is that oxidative stress in combination with loss of plasticizers/emulsifiers leads to an increased elasticity module with stiffness of the PP surface [37]. This may also lead to the rough and flaky surface described in many explanted meshes [5,6,8]. The known post crystallization effect and an evident polymorphism may be a reason for this phenomenon.

The hernia mesh in the abdominal wall is exposed to significant mechanical stress caused by biomechanics of the wall. The fibroblast–mesh interaction required for the integration of the mesh into the abdominal wall increases the stiffness of PP and triggers mechanical stress [38]. Totally disregarded within this concept remain the shearing forces working at the margins of the meshes at the junction to the native tissue. The increased elasticity modulus of PP due to the aforementioned leaching out of plasticizers together with the mechanically induced stress of the now recalcitrant surface would be such a hypothetical mechanism [9].

In addition to these changes in the human body, it often is neglected that the additives in the production process as well as the variabilities in the mesh knitting process may lead to specifications in the same material. It could be shown that during the knitting process, the linear density of the mesh is the principal factor affecting its mechanical properties, including resistance and flexibility [39,40]. It should be kept in mind that these concepts represent the beginning of the scientific analysis and are discussed very controversially.

### 4.4. Connection between Chemical and Physical Alteration Processes

Since preponderant superficial alterations of explanted PP materials are described, oxidative degradation is blamed as the trigger of these alterations. However, many questions remain open, for example: (i) is there a chance that the alterations were developed post explanation, (ii) are they just superficially and consequently of no relevance for the function of the PP mesh, or (iii) are chemical reactions with oxidative alterations of PP possible considering the local circumstance beneath the human abdominal wall?

With the here presented results, we were able to show that under very artificial conditions, these antioxidants and plasticizers are leached out of the polymer. In this context, two scenarios should be differentiated. (i) The first is thin samples of PP in which aging is not controlled by the diffusion process of oxygen. Degradation of the polymer is homogenous, and embrittlement results directly from structural changes. (ii) The second includes bulk samples in which aging takes place mainly at the superficial layer. Damage may result from density gradients [10]. Even if there are no homogenous thresholds for the differentiation between thin and bulky layers, we tend to favor variant (ii) for the fibers in our meshes.

It has been shown under in vitro conditions that oxidative stress seems to change the mechanical properties of PP meshes via chain entanglements and cross-linking [9].

Our results seem to facilitate the thesis that there are complex networks between chemical and physical processes. The oxidative stress dissolves the additives out of the PP; this led to an increase of elasticity modules presenting an embrittlement phenomenon on the surface of the polymer. Nevertheless, there is a lack of consensus regarding the mechanisms [9].

Compared with the aforementioned material PES is judged as more hydrophilic [3]. It remains unclear if the more often described alteration of some of these materials results from this more increased hydrophilicity [24,30].

### 4.5. Limitations of This Study

This study has several limitations: The evaluation emphasizes mainly NMR results, since with the other evaluation methods applied (e.g., infrared spectroscopy), no detection of differences was possible.

We did not perform photooxidation and thermal oxidation experiments, since 37.5 °C is for PP fibers a quite low temperature, but it represents the situation in the human body. Photooxidation is considered as a not relevant influence factor in the dark abdominal wall. Our experimental approach is very artificial. With this, we follow the same line as other approaches trying to analyze the process of alterations in alloplastic materials [10,11]. We consider the data shown here as a first step to investigate the possible dissolving of plasticizers out of alloplastic material more thoroughly and see this as a supporting argument for the assumption that the possible dissolving of antioxidants and plasticizers causes alterations in the physical properties of PP. A lack of antioxidants on the surface of the PP fiber may lead to an initiation of the autoxidation cycle by free radicals followed by chain scissions. The loss of plasticizers decreases its elasticity; the emerging tension on the material in combination with the mechanical strain of the abdominal wall can lead to the alteration of the PP.

## 5. Conclusions

Fifty years after the introduction of PP in daily surgical practice, it remains unclear why alterations in a usually inert alloplastic material occur. Chemical reactions causing a degradation of PP appear as an insufficient explanation.

We were able to show in our study that antioxidants and plasticizers may be dissolved from PP, whereas this was not the case in PES compounds. This elution of the antioxidants and plasticizers from the surface caused an increased elasticity modulus on the surface of the PP fiber, leading to superficial desquamations.

A mixture of chemical reactions (oxidative stress with additive leaching from polymer fibers) in association with physical alterations (increased elasticity modulus by loss of plasticizers) seems to be a source of these PP and PE alterations. This did not lead on to a complete dissolution of the fiber.

## Figures and Tables

**Figure 1 jfb-12-00024-f001:**
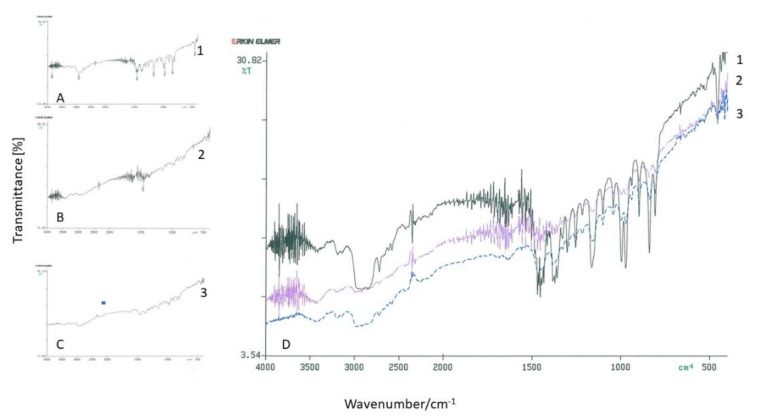
FT-IR spectra of Hertra 2© (1), Parietene 3© (2), and Prolene PMN 3© (3). (**A**–**C**) before incubation and (**D**) after incubation in hydrogen peroxide (representative experiment).

**Figure 2 jfb-12-00024-f002:**
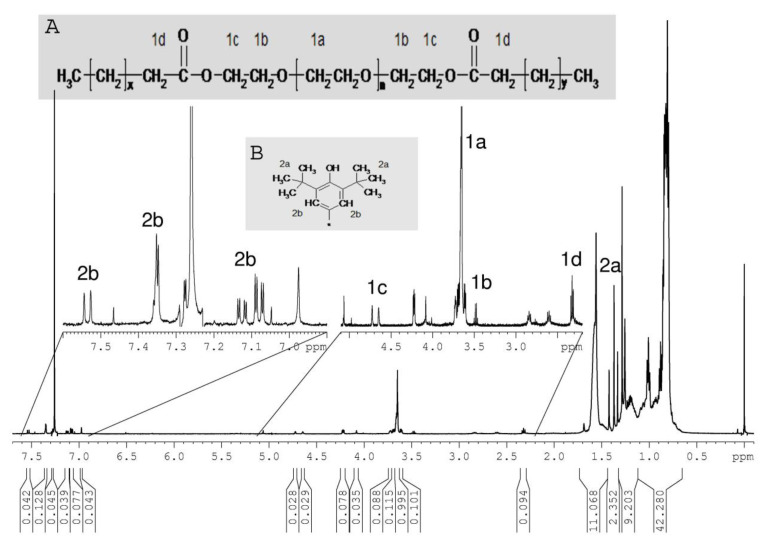
^1^H NMR spectrum of Hertra 2© chloroform extract (**A**): Chemical structure of PEG fatty acid ester additive, (**B**): Chemical structure of 2,6-di-tert-butyl phenol building block in antioxidants.

**Figure 3 jfb-12-00024-f003:**
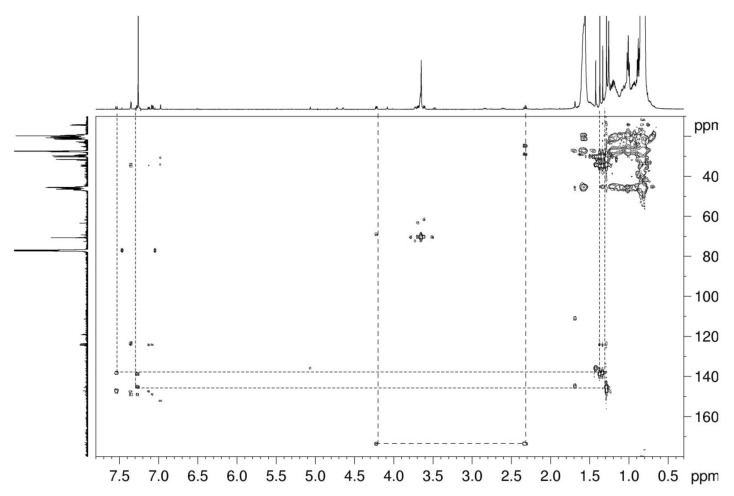
HMBC spectrum of Hertra 2©.

**Figure 4 jfb-12-00024-f004:**
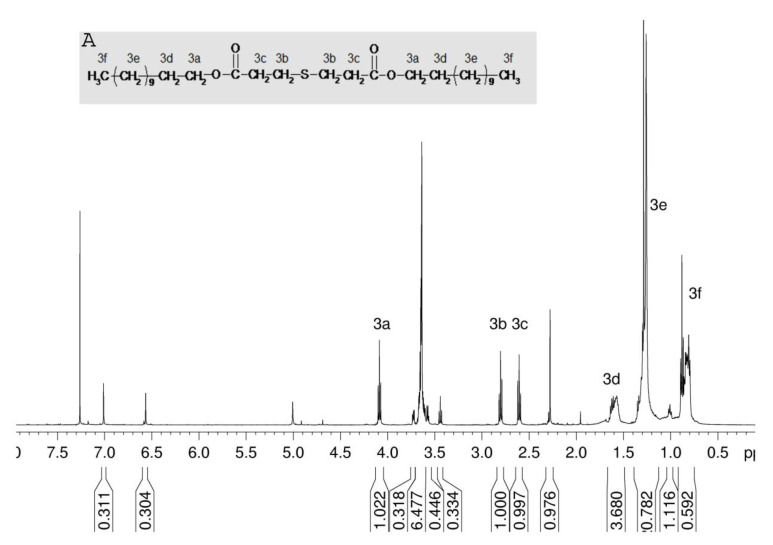
^1^H NMR spectrum of Prolene PMN 3© chloroform extract, (**A**): Chemical structure of didodecyl-3,3′-thiodipropionate auxiliary antioxidant.

**Figure 5 jfb-12-00024-f005:**
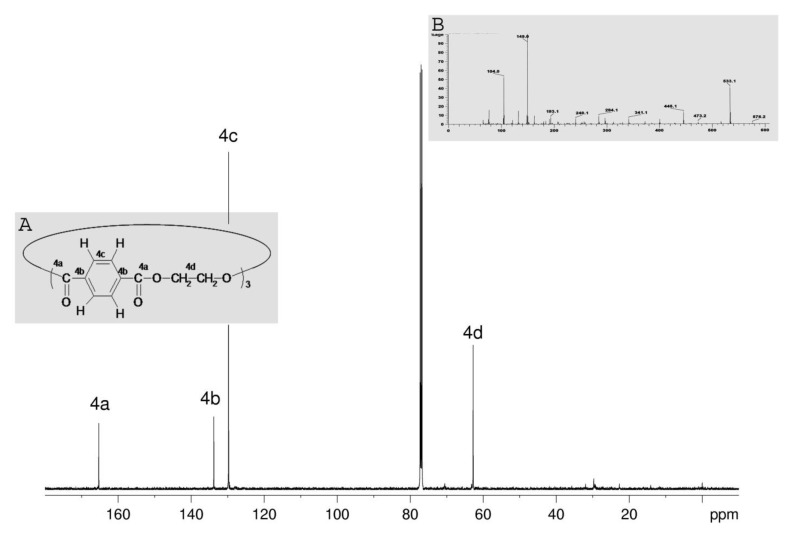
^13^C NMR spectrum of Parietex PES TEC 1510© chloroform extract, (**A**) Chemical structure of cyclic trimer of ethylene therephthalate, (**B**) Mass spectrum of Parietex PES TEC 1510© chloroform extract containing cyclic trimer of ethylene terephthalate.

**Table 1 jfb-12-00024-t001:** Investigated alloplastic materials.

Mesh	Manufacturer	Material	Filament
Hertra 2	Hernia mesh SRL Italy	PP Homopol.	Mono
Parietene3 PP1510	Medtronic/Covidien Germany	PP Homopol.	Mono
Prolene PMN 3	Ethicon Germany	PP Homopol.	Mono
Surgipro Mesh	Medtronic/Covidien Germany	PP Homopol.	Multi
Parietex PES TEC 1510	Medtronic/Covidien Germany	PE Copol.	Multi

PP = polypropylene, PE = polyester.

**Table 2 jfb-12-00024-t002:** Relative diameters of the various fibers after three weeks of incubation.

Mesh	Before Incubation	After Three Weeks				
		Aqua Bidest	Urea 50%	Saline Solution	Formalin 37%	H_2_O_2_
Hertra 2©	0.112 ± 0.004	0.109 ± 0.003	0.109 ± 0.003	0.109 ± 0.005	0.109 ± 0.005	0.109 ± 0.004
Parietene 3 PP 1510©	0.06 ± 0.000	0.06 ± 0.004	0.06 ± 0.007	0.06 ± 0.005	0.06 ± 0.005	0.006 ± 0.003
Prolene PMN 3©	0.08 ± 0.005	0.08 ± 0.004	0.08 ± 0.003	0.09 ± 0.005	0.09 ± 0.004	0.008 ± 0.004
Surgipro Mesh©	0.135 ± 0.019	0.137 ± 0.020	0.137 ± 0.018	0.141 ± 0.018	0.140 ± 0.020	0.135 ± 0.017
Parietex PES TEC 1510©	0.184 ± 0.012	0.182 ± 0.012	0.184 ± 0.011	0.188 ± 0.011	0.185 ± 0.012	0.185 ± 0.013

* Relative values, no units provided, *n* = 5 independent measurements *t*-test, *p*-value n.s.

## Data Availability

Data presented in this study are available on request from the corresponding author.

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
