# Peer review of "Increased Elasticity Modulus of Polymeric Materials Is a Source of Surface Alterations in the Human Body"

_jfb, 2021, doi:10.3390/jfb12020024_

Round 1

Reviewer 1 Report

The Authors have considered all questions and remarks, and essentially improved  the manuscript, beside a better presentation of the FTIR graphs. Anyway taking into account all Authors responds and the quality of the manuscript this paper may be published as it is.

Author Response

Reviewer 1:       No changes were recommended.

Reviewer 2 Report

This manuscript investigates if additives, supplemented to alloplastic mesh materials merge into the solution and become analyzable. The work is abundant and comprehensive and it’s worthy to publish.  However, the manuscript still needs some modifications.

It would be helpful if authors would consider about the following points:

  1. The words “may be” are not suitable in the title.
  2. The conclusion is encouraged to be enlarged so that the value of this work could be obtained clearly.

Author Response

- Words “may be” were changed in the title.

  • The conclusion section was enlarged. Please see line 351-367

Reviewer 3 Report

This revised manuscript present the degradation of alloplastic materials (surgical meshes) that are used as medical devices for hernia repair. The choice of partial dissolution of the polymers in chlorinated solvents is not correct mimicking of an in vivo oxidative condition (chloroform does not have oxygen), but the authors make clear the artificiality of the outcome in the description of the experimental design. The leaching of antioxidant compounds is not extensively demonstrated in water solutions but is demonstrated in chlorinated solvents. In this sense could serve as a proxy of a potential oxidation result in vivo (due to lack of antioxidants (Lines 345-347) and due to increase of elasticity modulus). Some minor points need to be addressed for the improvement of the manuscript before pubblication. A better proxy of a in body oxidation would be experiments of thermal oxidation or photooxidation, if this cannot be done please mention it in the discussion or perform thermo-oxidation experiments

Title: change module with modulus
Table 2: add units of length in the table
Line 94: how did the authors get 10-40 mg of chloroform-extractable materials after three weeks of dissolving the meshes in chloroform? was the chloroform evaporated after the three weeks and only 10-40 mg were recovered and used with deuterated chloroform for NMR characterisation?
Lines 103-104: this part is not clear, what do the authors refer to "with the fraction after thin layer chromatography", please specify further.
Line 337: remove "NMR pretty much" and change it with "mainly NMR results"

Author Response

- We did not perform thermal oxidation and photooxidation experiments. The explanatory statement we have written in line 339-342

  • “module” was changed in “modulus” in the title
  • We have explained in the Methods section (line 72-73), that since the absolute diameter of the fiber has no relevance for comparing the fiber before and after incubation, we did not calibrate our measurement system. So, the shown values are relative values and units are not possible to add.
  • The fact with the amount of recovered material was specified (line 94-96)
  • We considered TLC as standard procedure and would like to refer as an example to the following article: Thin-layer Chromatography-Nuclear Magnetic Resonance Spectroscopy – A versatile tool for Natural Products Analysis from Gössi et al (Chimia 2012, 66, No 5, page 347 ff). Here the procedure is described very detailed.
  • “NMR pretty much” was changed in “mainly NMR results”

This manuscript is a resubmission of an earlier submission. The following is a list of the peer review reports and author responses from that submission.

Round 1

Reviewer 1 Report

This manuscript investigates if additives, supplemented to alloplastic mesh materials merge into the solution and become analyzable. The work is abundant and comprehensive and it’s worthy to publish.  However, the manuscript needs some modifications so that it could be better than before.

It would be helpful if authors would consider about the following points:

  1. The title should be more accurate, rather than a style of reviews.
  2. The key points of the abstract are not obvious, and the logicality of the expression should be strengthened.
  3. The introduction is too short to illustrate the significance of this work, thus it should be enriched.
  4. Why do you choose 4 PP and 1 PES? Please explain this reason in the manuscript.
  5. “The measured thickness of the fibers after the three weeks incubation period did not significantly deviate compared to the material before incubation” (Line 96). What shall we know from this result?
  6. The section “discussion” is encouraged to be divided into related chapters of characteristics, so that the discussions could be more clearly and understandable.
  7. The conclusion is too brief to show the worth of this work.
  8. There are some grammatical errors, please check the paper carefully.
  9. The references nearly 5 years are very few, and the central references are too old.

Reviewer 2 Report

The paper by Kapischke M. et al, entitled “Alterations in polymeric alloplastic materials” presents characterization of four polypropylene and one polyester alloplastic material after incubation in different media in order to evaluated potential risk of additives leaching. Although the motivation of the work refer to the important issue in biomaterials presented work suffers from significant weaknesses.

Most important comments:

  1. Presented FTIR spectra are unacceptable quality. Baseline correction before data analysis should be performed. Even taking into account rough surface of analyzed samples, this step should be done before analysis. High noise signals observed near 4000 to 3600 cm-1 and 1800 to 1500 cm-1 could be results from the absorption of water vapor or carbon dioxide. What is more important, probably due to poor quality of obtained spectra no detailed description of characteristic peaks for each material is presented. The FTIR part strongly reduces the quality of the paper.
  2. The results related to NMR analysis are disorder and overinterprated. In line 129 suddenly Authors started to discuss the presence of signals characteristic for PEG and its derivatives with fatty acids without any basic analysis of all signals present on the spectra. This part is written in unclear and incomprehensible way. Additionally Authors doesn’t present any results related to the deuterated water extractions, mentioned H-H-COSY, HSQC analysis are also missing, so why it is mentioned in Materials and methods section? The good practice of leachable substances analysis is the use of proper reference substance. First of all the spectrum of such reference substance should be performed and then the extract should be analyzed or the reference should be added to the extracted solution and if an increase of characteristic signals is observed, direct confirmation of the presence of the substance is obtained.  
  3. Numerous mistakes in chemical compounds names are also present:
  • Line 16 and 53 should be hydrogen peroxide instead of hydrogen
  • Shortcut PES (line 27) is dedicated for poly(ether sulfone) polymer and could be not used for general name of polyesters. If the Parietex PES TEC 1510 is made from poly(ethylene terephthalate) (PET) why this name isn’t marked in table 1?
  • Line 66 – distilled water is not the same as deuterated water. It is unclear which water was used to obtain extracts, since shortcut in line 82 D2O suggest deuterated water.
  • Line 72 – from where distilled chloroform? Should be deuterated?

To summarize, poor results presentation quality and numerous mistakes in chemical terminology render the presented article unpublishable Journal of Functional Biomaterials and I recommend to reject this paper.

Reviewer 3 Report

Please find the attached document for specific remarks.

Reviewer 4 Report

This paper may be of a special interest in medical applications, particularly by surgeons and by specialist of biomedical engineering. The Authors have selected five commercial meshes used for the hernia surgery and checked experimentally the alternation of its surface after various treatment. From this point of view the paper, belonging to widely investigated and published implants as artificial materials used today, may be published. Prior to publications I’d like to advise to consider following question and queries.

  1. As mentioned above the commercially available materials have been selected for the experiments. Even the structure of the PP used in the meshes was not characterized. Polypropylene is a very general description of the polymeric material. Regarding its properties it may be a homopolymer or/and copolymer of PP widely used in various applications, thus it should be experimentally proved what kind of polymer is it, as the PP hompolymer and various copolymers of propylene may be very different regarding structure and properties.  You may find a huge bibliography explaining the structure-properties relationship of PP and its copolymers.
  2. The introduction should be better presented. The literature sources are just named and not specified. The bibliography of the medical application of polyolefins is missing.
  3. A certain incubation induced changes of the average diameter value were observed; at least a brief explication is missing why a decrease (and not an increase) of the diameter was observed for some samples.
  4. The FTIR spectra (Fig. 1) are of a very poor quality; therefore the discussion of the results is not evident; but if look carefully at the spectra changes of the maxima existence and disappearance or intensity changes may be seen. Pls give comments.
  5. Lines 261-262; what do you mean under “loss of elasticity” a higher or lower elasticity modulus? A short description how these values were measured is missing. It’s well known that PP is a semi-crystalline copolymer presenting both a strong post crystallization effect and an evident polymorphism. Don’t you think that these two effects may be the origin of mechanical changes as it may be found in many papers?
  6. Line 263 you mention that the “mechanical induced stress” may appear, the question is – due to what? shrinkage? or other?
  7. In line 270 the “oxidation stress” leading to entanglements and cross-linking is suggested; pls comment if you have any proof of both effects like macromolecular chains entanglements and cross-linking of the PP.
  8. Lines 265-268; the composition of the polymeric material is not known, the proposed leaching of these additives should be better described and experimentally proved, as practically only the presence of PEG as additive was proven.
  9. The changes of physical properties are polymer structure dependent, on the contrary the surface properties are related to skin layer modification; both effects are partially “mixed together” in your explanation; pls explain both phenomena separately to help the reader to understand your investigation and analysis.
  10. Line 290; what is your idea about? "The mechanical strain or stresses are leading to the alternation of PP", what is the relationship between stress and/or strain and alternation of the polymer?